# Ultra-Low-Pressure Membrane Filtration for Simultaneous Recovery of Detergent and Water from Laundry Wastewater

**DOI:** 10.3390/membranes12060591

**Published:** 2022-06-01

**Authors:** Yusran Khery, Sonia Ely Daniar, Normi Izati Mat Nawi, Muhammad Roil Bilad, Yusuf Wibisono, Baiq Asma Nufida, Ahmadi Ahmadi, Juhana Jaafar, Nurul Huda, Rovina Kobun

**Affiliations:** 1Faculty of Applied Science and Engineering, Universitas Pendidikan Mandalika UNDIKMA, Jl. Pemuda No. 59A, Mataram 83126, Indonesia; yusrankhery@gmail.com (Y.K.); soniaelydaniar@gmail.com (S.E.D.); baiqasmanufida@undikma.ac.id (B.A.N.); ahmadi_kim@yahoo.co.id (A.A.); 2Department of Chemical Engineering, Universiti Teknologi PETRONAS, Bandar Seri Iskandar 32610, Perak, Malaysia; normi_16000457@utp.edu.my; 3Faculty of Integrated Technologies, Universiti Brunei Darussalam, Gadong BE1410, Brunei; 4Department of Bioprocess Engineering, Brawijaya University, Jl. Veteran, Malang 65145, Indonesia; y_wibisono@ub.ac.id; 5Advanced Membrane Technology Research Centre (AMTEC), School of Chemical and Energy Engineering, Universiti Teknologi Malaysia, Skudai 81310, Johor, Malaysia; juhana@petroleum.utm.my; 6Faculty of Food Science and Nutrition, Universiti Malaysia Sabah, Kota Kinabalu 88400, Sabah, Malaysia; rovinaruby@ums.edu.my

**Keywords:** gravity-driven membrane filtration, laundry wastewater, ultra-low-pressure filtration, membrane fouling, ultrafiltration

## Abstract

Reusing water and excess detergent from the laundry industry has become an attractive method to combat water shortages. Membrane filtration is considered an advanced technique and highly attractive due to its excellent advantages. However, the conventional membrane filtration method suffers from membrane fouling, which restricts its performance and diminishes its economic viability. This study assesses the preliminary performance of submerged, gravity-driven membrane filtration—under ultra-low trans-membrane pressure (△*P*) of <0.1 bar—to combat membrane fouling issues for detergent and water recovery from laundry wastewater. The results show that even under ultra-low pressure, the membrane suffered from compaction that lowered its permeability by 14% under △*P* of 6 and 10 kPa, with corresponding permeabilities of 2085 ± 259 and 1791 ± 42 L/(m^2^ h bar). Filtration of a detergent solution also led to up to 8% permeability loss due to membrane fouling. During the filtration of laundry wastewater, 80–91% permeability loss was observed, leading to the lowest flux of 15.6 L/(m^2^·h) at △*P* of 10 kPa, 38% lower than △*P* of 6 kPa (of 25.2 L/(m^2^·h)). High △*P* led to both the membrane and the foulant compaction inflating the filtration resistance. The system could recover 83.6% of excess residual detergent, while most micelles were rejected (ascribed from 71% of COD removal). The TDS content could not be retained, disallowing maximum resource recovery. A gravity-driven filtration system can be self-sustained with minimum supervision in residential and industrial laundries. Nevertheless, a detailed study on long-term filtration performance and multiple cleaning cycles is still required in the future.

## 1. Introduction

The water scarcity issue can be addressed through efficient water reuse and recycling, thereby reducing the need for and load of wastewater treatment before the water is discharged into water bodies and then treating again for usage [1,2]. Reusing water and detergent from the laundry industry is an attractive method to combat water shortage; it increases economic viability and protects the environment due to the detrimental impact of excess waste detergent [3]. Some surfactants may contain toxic components and promote the growth of some bacterial populations that are more sensitive to laundry wash toxins [4,5]. Industrial laundry requires 15 L/kg of freshwater to wash process garments [6], while hotels and hospitals consume at least 24 L of freshwater per occupied room a night for laundering [1]. Meanwhile, 60–140 L of freshwater is used per washing cycle in residential laundries [7]. Laundry wastewater from small-scale laundry services and residential settings is typically discharged as greywater, containing detergent that poses detrimental environmental impacts. Surfactants can accumulate in organisms and cause eutrophication of aquatic environments, and some are resilient to biodegradation [8,9].

Table 1 summarizes recent research on detergent recovery from laundry wastewater using membrane filtration. Membrane filtration is considered an advanced technique and is highly attractive due to its low energy usage, its small footprint, and the small number of chemicals that are required [10]. However, most previous studies developed a treatment process to remove detergent altogether from the treated water [11]. Consequently, the treatment often consisted of other processes to accompany membrane filtration, i.e., coagulation and flocculation as pre-treatment methods to boost the treatment’s performance. Treatment and purification of laundry wastewater for water reuse could be achieved by integrating crossflow ultrafiltration (UF) with other units. These include physico-chemical pre-treatments, such as sand filtration, ozonation, and granular activated carbon adsorption [6]. The array of those four processes removed the chemical oxygen demand (COD), total suspended solids (TSS), and turbidity removal efficiencies of 87, 98, and 99%, respectively. The UF was used as the post-treatment unit to eliminate the residual pollutant—including the excess detergent—and allowed effluent reuse for washing. Despite applying four separation units, the system aimed only to recover and reuse water and eliminate the excess detergent. A combination of a membrane bioreactor and nanofiltration was deemed suitable for the treatment and recycling of laundry wastewater to achieve a permeate COD of <50 mg/L and an anionic surfactant concentration of <0.5 mg/L [10]. In other studies, coagulation [12] and ozonation [13] were used as pre-treatments to minimize membrane fouling, achieving permeability of 160–450 L/(m^2^ h bar) and 25 L/(m^2^ h bar), respectively.

Standalone membrane filtration systems have also long been used for laundry wastewater filtration, including the recovery of detergent compounds [14]. A cross-flow UF using a lab-made polyethersulfone showed a low permeability of 11.04 Lm^−2^h^−1^bar^−1^ obtained under △*P* of 5 bar due to cake layer fouling. In another study, increasing △*P* from 0.5 to 1.5 bars enhanced the cake formation on the membrane surface and contributed to membrane fouling [15]. This suggests that the application of high △*P* caused the foulant layer to be more compact and increased the overall filtration resistance. 

Most of the previous research has focused on developing or integrating processes for laundry wastewater treatment with the aim of producing an effluent that meets the discharge standard and water reuse specifications. However, such approaches are costly and have a large footprint. However, when a standalone membrane was applied, it suffered from severe membrane fouling, resulting in low water fluxes and requiring a large membrane area to meet the required treatment capacity or application of high △*P*. Typical △*P* values for membrane filtration operation were 0.1–1.0 bar with water fluxes of 50–100 Lm^−2^h^−1^ [16]. 

The application of high △*P* was implemented to compensate for low permeability due to membrane fouling. Increasing △*P* led to higher flux, hence reducing the membrane footprint. However, high △*P* in a crossflow filtration system has been associated with severe membrane fouling, as revealed in previous studies [15,21], and thus can lead to high pumping energy and complex cleaning operation. Therefore, a new approach of using a standalone, gravity-driven, ultra-low pressure (ULP) membrane filtration system (△*P* < 0.1 bar) with the aim of detergent and water recovery has been investigated [17,18]. Aiming for detergent recovery excluded the need for a complex operation. Gravity-driven membrane filtration uses hydrostatic pressure to drive the filtration, with the typical operating pressures of 0.4–0.1 bar [24], equivalent to 40–100 cm water heads. The method has been successfully applied for water and wastewater treatment, achieving low but sustainable flux without physical or chemical cleaning [25]. The attainment of a stable flux was attributed to the presence of a biofilm that controls the filtration resistance.

Nonetheless, the application of ULP with low fluxes also resulted in a low drag force that typically brings the foulant to the pore mouth. Our previous work employed a submerged vacuum filtration system to treat laundry wastewater for detergent and water reuse purposes [17,18]. Operation under a vacuum would require extra pumping energy to hinder the self-operation system that could otherwise be achieved using a gravity-driven membrane filtration system. In addition, aeration was also implemented to control membrane fouling, which could further complicate the operation.

This study assesses ULP submerged membrane filtration to treat real laundry wastewater for detergent and water recovery. A gravity-driven system from a feed water head was implemented to create a self-operating system with minimal energy input. Firstly, the impact of △*P* on clean water permeability was evaluated. Then, filtrations of a detergent solution (detergent + water) and laundry wastewater were conducted. The permeation, rejection, and detergent recovery performances of the system were then evaluated. Finally, a conceptual process for implementing the system in a small-scale industrial or residential laundry was also designed.

## 2. Materials and Methods

### 2.1. Laundry Wastewater and Analytical Methods

Three samples were collected and analyzed, namely, (1) the mixture of the detergent and water (before washing), (2) the wastewater after washing, and (3) the permeate solution. The detergent was obtained from a local supplier (CV Chemica Karya, Mataram, Indonesia). For the first sampling point (before washing), an 8 gL^−1^ detergent solution (dissolved in tap water) was prepared as suggested by the supplier. For the second sampling point, real laundry wastewater (after the washing and rinsing stage) was collected from a local residential laundry and was used as the feed for the filtration test. The properties of the real laundry wastewater and the analytical methods are shown in Table 2.

### 2.2. Gravity-Driven Filtration Set-Up

The filtration performance was evaluated using a gravity-driven, constant-pressure filtration set-up, as illustrated in Figure 1. The △*P*s were set at 6, 8, and 10 kPa, corresponding to 50, 80, and 100 cm water heads, respectively, controlled by the feed levels. The level was maintained by recirculation via overflow at the designated feedwater level. The permeate was collected at the bottom of the filtration tank and—after volume measurement—was returned to the feed tank to maintain the constant feed condition. The permeate was collected and measured semi-batch-wise every 5 min of filtration. The set-up was equipped with a u-shaped hollow fiber polyacrylonitrile ultrafiltration membrane with a nominal pore size of 0.01 µm. The feed was recirculated at a rate of 1.5 L/min.

### 2.3. Filtration Test

Three filtration tests were performed using different feeds: clean water, detergent solution, and laundry wastewater. The filtration tests were conducted under three △*P* of 6, 8, and 10 kPa, corresponding to 60, 80, and 100 cm feed water heads, respectively. The tests were run in triplicates. The permeate flux (J, L/(m^2^ h)), permeability (L,  L/(m^2^ h bar)), and rejection (*R*, %) were calculated using Equations (1)–(3), respectively.
(1)$$J=\frac{V}{t\;A}$$
(2)$$L=\frac{J}{\Delta P}$$
(3)$$R=\frac{C_{F}-C_{P}}{C_{F}}\times 100\%\;\;$$
where V is the volume of the collected permeate (*L*), A is the membrane surface area (0.242 m^2^) t is the time taken to collect the permeate (h), ΔP is the applied transmembrane pressure (bar), $C_{F}$ is the concentration in the feed (g/L), and $C_{P}$ is the concentration in the permeate (g/L).

## 3. Results and Discussion

### 3.1. Effect of Pressure on Clean Water Permeability

Figure 2 shows the effect of △*P* on clean water permeability and flux, evaluated at 6, 8, and 10 kPa. It shows a slight decline in permeability over time, which stabilizes toward the end of the experiment. Additional testing, extending the filtration time to six hours, did not significantly change the final permeability value. The decrease in permeability at higher △*P*s demonstrates the prominence of membrane compaction (Figure 2B), even under ULP, which is rarely reported in the literature. The applied physical pressure compression of the membrane structure increased the intrinsic filtration resistance [26]. The permeability decreased significantly from 2085 ± 259 L/(m^2^ h bar) under TMP of 6 kPa to 1791 ± 423 L/(m^2^ h bar) under a △*P* of 10 kPa, which is attributed to the state hydration of the membrane as the water is being forced out from the membrane matrix at higher △*P*s [27].

The finding on permeability, shown in Figure 2, suggests the importance of the applied △*P* on permeability as directly affecting membrane compaction. It is known that membrane compaction might reduce the membrane pore size or lead to the deformation of pore geometry [28]. The findings justify the application of the ULP filtration system, which offers the benefit of lower intrinsic membrane resistance due to a lower degree of compaction under a low △*P*, and are consistent with a recent report [29]. Figure 2 shows that the increase in △*P* of 66.7% from 6 to 10 kPa corresponded to a smaller increase in water flux of 43.1% from 125 ± 16 to 179 ± 4 L/(m^2^ h). Our earlier work [17] reported a similar finding when applying ULP for detergent wastewater filtration to avoid permeance loss due to membrane compaction at high △*P*.

It is worth noting that membrane compaction was highly reversible judging from the experiment repetitions with low variability in multiple tests. The tests were undertaken from 6 to 10 kPa and were repeated twice but still resulted in similar permeabilities, suggesting the reversible nature of the compaction. The %stdev values of repeated filtration tests were only 2–12%. The occurrence of membrane compaction was almost instantaneous (within less than 5 min) at the beginning of filtration, as shown by sudden drops of clean water on the first data point for each tested △*P*, followed by a slight decline in permeability over the extended filtration time.

### 3.2. Detergent Solution and Laundry Wastewater Filtration

Figure 3 shows the permeabilities of detergent solution tested under different △*P*s of 6, 8, and 10 kPa. The permeability trend and the effects of the △*P*s were similar to those of clean water filtration. However, a slight degree of membrane fouling was observed. The permeabilities of detergent solution evaluated at △*P*s of 6, 8, and 10 kPa were 1927 ± 24, 1793 ± 23, 1709 ± 1 L/(m^2^ h bar), 8, 3, and 5% lower than the clean water permeability, respectively. This finding suggests that detergent in the feed solution had a minor impact on membrane fouling. Despite foulant deposition on the membrane surface, the membrane could maintain its performance throughout the 30 min of filtration due to the ultra-low pressure applied to the system, which minimized the fouling rate.

Figure 4A shows the permeability of laundry wastewater filtration under various △*P*s as a function of filtration time. The final permeability values were 04.2 ± 4.1, 330.2 ± 2.6, and 153.7 ± 1.1 L/(m^2^ h bar) for △*P*s of 6, 8, and 10 kPa (Figure 4B), respectively. The obtained permeabilities in this study are much higher than those reported in the literature (see data in Table 1), suggesting that lower △*P* led to less membrane compaction and was less prone to membrane fouling. It is worth noting that the reported permeabilities did not reach a steady state due to the short duration of the filtration test; hence, a slow decline was still expected when the filtration was prolonged.

Interesting results are shown for the flux data. They are contrary to those of prior studies [21,30,31]. Increasing the filtration driving force (△*P*s) by raising the feed’s hydrostatic pressure lowered the flux from 25.2 ± 0.2 to 15.6 ± 0.1 L/(m^2^ h) due to 80 and 91% of membrane fouling for △*P*s of 6 kPa and 10 kPa, respectively. The higher membrane fouling rate for the latter might be due to the higher impact of foulant accumulation on the membrane surface, which sped up the cake layer formation, thus reducing permeability performance [17,29]. This finding also suggests the possibility of operating laundry wastewater filtration under ULP, which is beneficial for enhancing throughput and maintaining filterability performance, as demonstrated by the results in this study.

The initial reading of permeability, using wastewater as the feed (Figure 4A), was much lower than the final value of clean water permeability (see Figure 2). This implies that high water transportation across the membrane pores, leading to the very high initial flux at the beginning of the operation, imposed the drag force that carried the foulant material to the pore mouth and rapidly fouled the membrane, which is in accordance with the findings of others [17,30,31,32,33]. The drag force was more prominent than the back diffusion of the foulant away from the membrane, which worsened the membrane performance due to fouling. As the flux decreased, it lowered the force dragging the foulant toward the pore mouth and slowed down the foulant accumulation.

It should be noted that this study was performed by assuming the typical temperature of laundry wastewater to range from 25 to 40 °C [34], as discharged from “cold” or “warm” washing options. Thus, the proposed implementation might be less suitable for laundry wastewater with temperatures >80 °C. High temperatures might deteriorate the lifespan of the polyacrylonitrile ultrafiltration membrane. However, warm temperatures can promote permeation by lowering feed water viscosity and reducing the propensity of irreversible fouling [35].

### 3.3. Rejection Performance and Detergent Recovery Rate

Figure 5 shows the membrane’s rejection of detergent, COD, color, TP, and TDSs. The initial values for the feed laundry wastewater are provided in Table 2. It can be observed that the applied UF membrane could recover 16.4% of residual excess detergent that was still present in the wastewater. This proves the hypothesis that the residual detergent could still be recovered in the permeate stream for reuse without degradation/elimination for discharge purposes by implementing the multiple treatment units listed in Table 1. It was speculated that the bound—or consumed—detergent molecules attached to a large substrate that was retained by the membrane pores. The formation of a dynamic foulant layer atop the membrane surface also helped to enhance the detergent rejection.

The membrane retained 71% of COD, which measures organic matter and other reducing substances [21]. The laundry wastewater feed had a high COD content of 1060 mg/L, likely originating from anionic surfactants, builders, and other oil substances [22]. Most of it (71%) was retained by the membrane. It is postulated that the detergent was not entirely used up during the washing cycle. Laundry wastewater is comprised of detergent in micelles and some in free form. The latter can be considered as the excess detergent unused for washing purposes. Therefore, the reuse of permeate water from this filtration would be implicated in the reuse of excess detergent.

As shown in Figure 5, the system also shows a high removal of TP (93.6%). TP probably bonded to the organic molecules to form a large aggregate, making it easier to remove using membrane filtration with UF properties. Phosphate is present in most detergent formulations as a sequestering agent to reduce water hardness, as it can interfere with detergent activities [36]. Since high phosphate content in wastewater may lead to eutrophication in water bodies, this finding (high TP removal) is favorable for managing laundry wastewater [37].

Indeed, the membrane could not reject the dissolved particles because of the ions in the solution that were relatively much smaller than the membrane’s nominal pore size. The poor rejection of TDSs may result in their build-up in multiple reuses of the detergent and water, which becomes a long-term issue for implementing the proposed concept.

The detergent rejection rate was 16.4%, suggesting that most of the detergent presented in the laundry wastewater passed with the permeate water (83.6%). The poor detergent rejection was most likely due to the relatively large size of the free-detergent molecule when compared with the membrane pore size. The detergent was expected to be bound to dirt.

### 3.4. Practical Implementation

Figure 6 illustrates a block flow diagram displaying the implementation of gravity-driven membrane filtration integrated into the current industrial or residential washing system. The proposed concept complies well with the existing operation, to which an additional two tanks can be installed. Implementing the idea of partial reuse of laundry wastewater for washing garments involves make-up detergent and freshwater. The wastewater generated during the washing is filtered to a filtration tank equipped with submerged membrane filtration. Preferably, the filtration tank is at a lower level than the effluent discharge of the washing machine, so the wastewater flows under gravity. The permeate line is linked to a reuse water storage tank that stores the permeate before being reused for the next washing cycle. Some of the wastewater could be blown down to avoid TDS accumulation and to channel the overflow of wastewater that the membrane filtration could not recover. Some blowdown water is mixed with freshwater fed to the reused water storage, where the feed water for washing is pumped. The used detergent that could not be recovered also needs to be made up for. Such a system can function with minimum supervision and is very practical for implementation in small-scale industrial laundry. This proposed system is much simpler than our earlier proposal [17], in which the system was run under vacuum (requiring permeate pumping) and with aeration as the means of membrane fouling control (requiring air compressor). This system may require regular membrane cleaning, such as backwashing, to reverse the fouling effect by flushing the permeate through the membrane to the feed side, achieving a high shear rate and lifting off the deposited foulant from the membrane surface. Other than mitigating fouling, membrane cleaning may prolong the lifespan and functionality of the membrane [38].

Water and laundry reuse from laundry wastewater could substantially reduce the volume of water and detergents going to waste. Employing ultrafiltration was estimated to achieve a 40% recovery of reusable surfactants by recycling laundry wastewater [39]. On a larger scale, recovered detergent could be used in lower grade applications, such as road cleaning and car washing, as suggested earlier [10]. However, the implementation of such an idea may be hindered by logistic/transportation issues.

## 4. Conclusions

This study assessed the performance of ULP gravity-driven ultrafiltration for excess detergent and water recovery from laundry wastewater. It was found that the membrane suffered compaction that lowered its permeability when operated at high △*P*. Up to 14% clean water permeability losses occurred when increasing the △*P* from 6 kPa to 10 kPa with corresponding permeability values of 2,085±259 and 1791±42 376 L/(m^2^ h bar), respectively. Despite the compaction, the ULP resulted in substantially high permeability. The membrane also suffered permeability losses by up to 8% when treating detergent solution. When treating the real laundry wastewater, membrane fouling led to 80–91% permeability loss, suffered mainly by the system with the highest △*P* of 10 kPa. As a consequence, it showed the lowest flux of 15.6 L/(m^2^ h), 38% lower than the one with the lowest driving force of 6 kPa (of 25.2 L/(m^2^ h)). High △*P* led to foulant compaction, and consolidation resulted in higher filtration resistance. The system could recover 83.6% of excess residual detergent, and most of the micelles could be removed from the permeate. However, no retention of TDS was achieved, implying the need for blowdown or partial recovery. The proposed method can be practically implemented and self-sustained with minimum supervision. Nevertheless, a detailed study on reusability and long-term performance of the system is still required in the future.

## Figures and Tables

**Figure 1 membranes-12-00591-f001:**
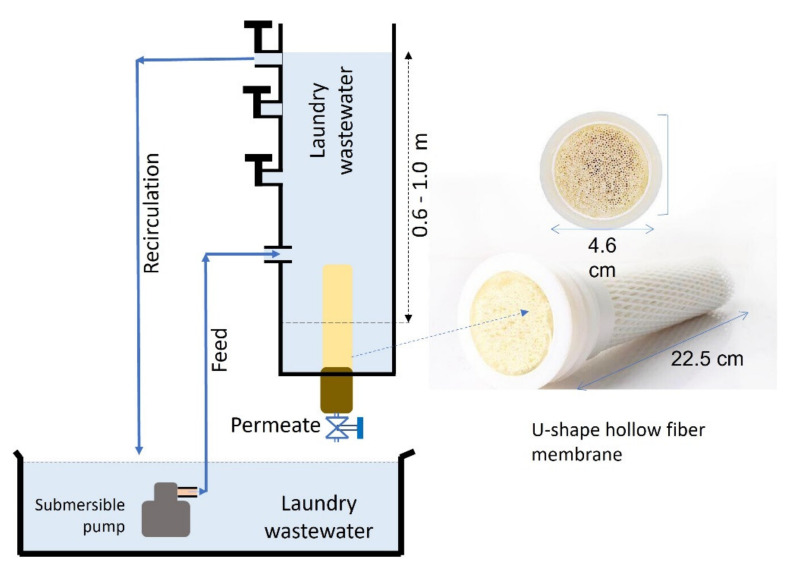
Illustration of gravity-driven, submerged membrane filtration set-up showing the feed overflow recirculation system that is used to maintain the feed level and the u-shaped hollow fiber membrane installed at the bottom of the filtration tank.

**Figure 2 membranes-12-00591-f002:**
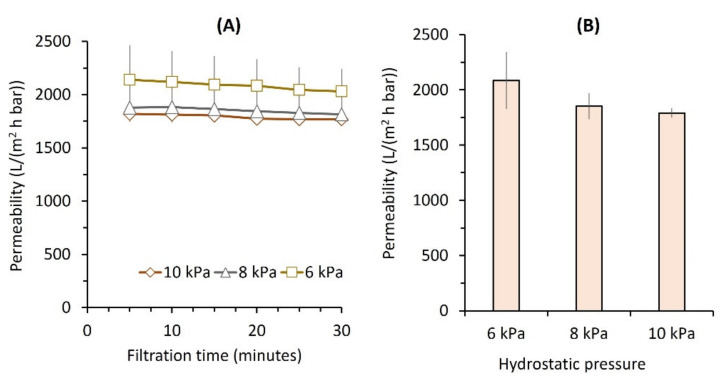
Clean water permeability as a function of applied pressure shown (**A**) over time and (**B**) the final values.

**Figure 3 membranes-12-00591-f003:**
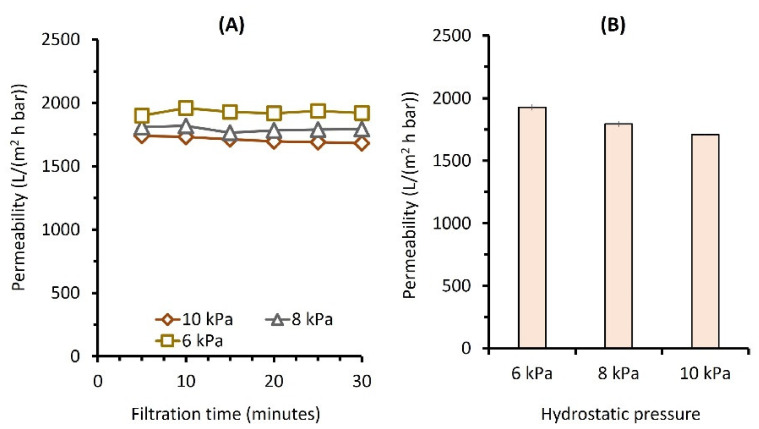
Filtration of detergent solution as a function of applied pressure shown (**A**) over time and (**B**) the final values.

**Figure 4 membranes-12-00591-f004:**
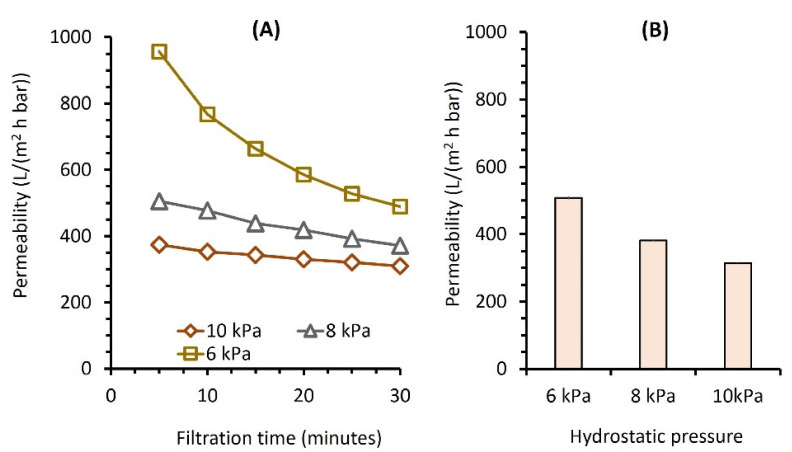
Filtration of laundry wastewater as a function of applied pressure shown (**A**) over time and (**B**) the final values.

**Figure 5 membranes-12-00591-f005:**
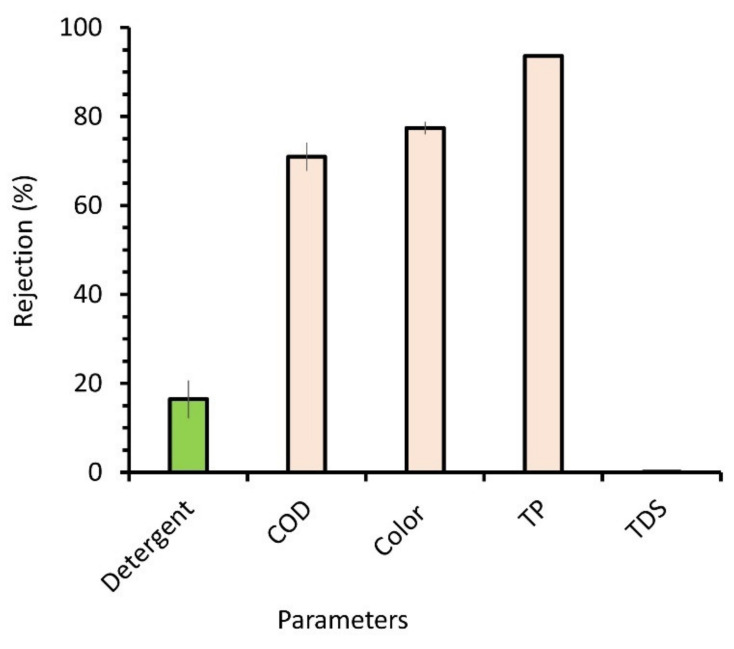
Rejection performance of laundry wastewater filtration.

**Figure 6 membranes-12-00591-f006:**
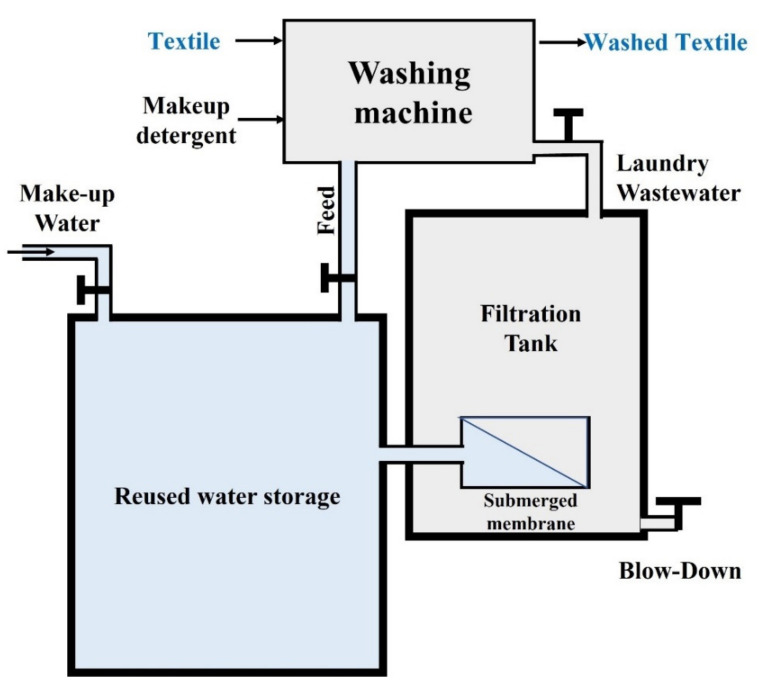
Practical implementation of reuse of laundry wastewater for laundry services.

**Table 1 membranes-12-00591-t001:** Recent studies on the treatment of laundry wastewater for water and/or detergent reuse.

System/Source of Wastewater	Purpose of Study	Pressure (kPa)	Permeability (Lm^−2^h^−1^bar^−1^)	Removal (%)					Ref.
				COD	Color	TDSs	NTU	TP	
UF/residential laundry	Detergent and water recovery	Vacuum of <10	150–297	52	-	-	97.9	65.3	[17]
UF/residential laundry	Detergent and water recovery	10	100–200	57	-	-	77.0	30.0	[18]
MF/residential laundry	Water recycle	50	18.72	5.5	-	2.5	98.4	-	[7]
Physico-chemical pre-treatment, sand filtration, ozonation, GAC filtration and UF/domestic laundry	Water reuse	-	84.0	87.0	-	-	99.0	-	[6]
Combined coagulation, flocculation, sedimentation, and MF or UF/industrial laundry	Wastewater treatment	140	12.5–92.2	68.8	98.4	55.0	99.1	-	[19]
Combined coagulation, flocculation, sedimentation, adsorption, MF/industrial laundry	Wastewater treatment	140	43.2	80.0	99.9	-	99.4	-	[20]
UF/laundry center	Wastewater treatment	100 600	25.0 11.04	88.0	-	82.0	98.0	-	[21]
Coagulation + MF/industrial laundry	Wastewater treatment	70	160–450	65.0	-	-	100.0	-	[12]
UF/hospital laundry	Effluent treatment	300–500	30–50	53.6	-	-	-	95.4	[22]
RO/hospital laundry	Effluent treatment	300–500	7.4–12.3	98.9	-	-	-	98.6	
Ozone + UF/domestic laundry	Water recycle	40	25.0	95.0	-	-	-	-	[13]
MF ceramic membrane/domestic laundry	Water recycle	300	30.0	80.0	-	-	95.0	-	[23]
UF ceramic membrane/domestic laundry	Water recycle	300	16.7	83.8	-	-	99.5	-	
UF/domestic laundry	Detergent and water recovery	6	500.4	71.0	78.0	-	-	93.6	This study

GAC: granular activated carbon, COD: chemical oxygen demand, TDSs: total dissolved solids, NTU: nephelometric turbidity units, UF: ultrafiltration, MF: microfiltration, RO: reverse osmosis.

**Table 2 membranes-12-00591-t002:** Properties of the laundry wastewater and standard methods for their determination.

Parameters	Unit	Value	Analytical Method
pH	-	7.9	SNI 06-6989.11
Detergent	(mg/L)	2.56	SNI 06-6989.51
COD ^1^	(mg/L)	1060	SNI 6989.2
Color	(Pt.Co)	229.8	SNI 06-6989.80
TP ^2^	(mg/L)	1.271	SNI 06-6989.31
TDSs ^3^	(mg/L)	692.6	APHA 2540 C

^1^ Chemical oxygen demand; ^2^ total phosphorus; ^3^ total dissolved solids.

## Data Availability

Not applicable.

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
