# Peer review of "Ultra-Low-Pressure Membrane Filtration for Simultaneous Recovery of Detergent and Water from Laundry Wastewater"

_membranes, 2022, doi:10.3390/membranes12060591_

Round 1

Reviewer 1 Report

In the file provided I am unable to see clearly where these changes have been made. There are no references to lines numbers changed and detailed description of the changes made. I recommend that a new manuscript with the changes clearly visible should be prepared (changes could be in a different color to highlight them, for example). See embedded comments (in italics) below: 

 Reviewer #2

Comment 1: This paper deals with a an important issue of water reuse in laundry operations, using simple low-tech solutions. The writing and grammar is reasonable but there are places were an English language edit is necessary, eg. Abstract, line 211.

Response to comment 1: As pointed out by the reviewer, the abstract and mentioned line (Line 211) has been revised accordingly.

This is insufficient. “Line 211” was just one example of where editing is required. A thorough review/edit of the entire manuscript is necessary.

Comment 2: The introduction is sufficient although there is some unnecessary repetition (lines 83-88 could be cut/condensed).

Response to comment 2: As suggested by the reviewer, the highlighted section has been revised accordingly.

Provide detail on how the manmuscript has been revised. Be specific in outlining your changes.

Comment 3: The methods are well-explained and thorough, although would be improved by adding the water quality characteristics of the “tap water”.

Response to comment 3: Thank you for the suggestion, however, we believe that the characteristics of tap water is unnecessary for this study since we already prepared the detergent in water solution to be compared with wastewater.

I do not understand how the reply addresses the comment. Please revise language for clarity.

Comment 4: Comment should be added to explain the influence and significance of “unsteady state” concentration in the wastewater reservoir for the experiments. For example, the volume of the reservoir and recirculation/feed flowrates should be noted, and the impact of changing concentration (in the reservoir) on the fundamental equations (1-3) should be explained (i.e. do these equations still hold given varying concentration?).

Response to comment 4: The permeate was collected and measured semi-batch-wise for every 5 mins of filtration, and it was returned to the feed tank to maintain the constant feed condition, including feed level and concentration. This information is available in the manuscript (Section 2.2).

The response here discusses sampling methods rather than the fundamental issue of the relevance of the defining equations. Please address the issue.

Comment 5: Unclear symbols at line 158: Cf and Cp ?

Response to comment 5: Cf is the concentration of feed while Cp is the concentration of permeate. The symbols and definitions have been amended accordingly.

Ok.

Comment 6: Section 3.2, lines 196-200. I am unconvinced of the assumed influence of fouling ? If that were the case then explain why this effect would not increase over time.

Response to comment 6: We believe that the presence of detergent in the feed solution contributes to the membrane fouling. However, the ultra-low pressure applied in this system is able to reduce the fouling rate and minimize the impact of fouling throughout the 30-min filtration process. The justification has been added to the manuscript.

I remain unconvinced of the reasoning. Perhaps outlining here in detail what changes were made to the manuscript might help.  

Comment 7: Lines 214-218: this paragraph is confusing. More extensive reporting of flux data would help. As such, paragraph at lines 220-226 feels like conjecture. Evidence should be presented. 2

Response to comment 7: The writing in the highlighted paragraphs have been amended and more justification/evidence has been added to the revised manuscript.

Outline here in detail what changes were made to the manuscript.  

Comment 8: Line 260-261: I found this result surprising? What have others observed with respect to TDS ? Relate your results back to existing/previous literature.

Response to comment 8: The data of TDS retention of other studies from literatures are now available in Table 1. Most of previous study did not measure the TDS rejection, and when measured it ranged from 2.5 to 88%, which likely affected by the membrane type and the applied process. In this case, the applied UF membrane could not reject the TDS and its consequences has also been discussed in the manuscript.

Outline here in detail what changes were made to the manuscript.  

Comment 9: Discussion and conclusion: Your paper would benefit from a comment on how long the membrane will last and how it might be cleaned, to provide practical advice on implementation.

Response to comment 9: Thank you for the valuable suggestion. The idea of membrane cleaning has now been added to the revised manuscript. However, a detailed long-term filtration study would be required to estimate the lifespan of the me

Sounds ok but outline here in detail what changes were made to the manuscript.  

Reviewer 2 Report

Thank you for inviting me to review the manuscript titled “Ultra-Low-Pressure Membrane Filtration for Simultaneous Recovery of Detergent and Water from Laundry Wastewater”. The research aim is to assess the performance of a submerged gravity-driven membrane filtration under ultra-low trans-membrane pressure (â–³P) of <0.1 bar to combat membrane fouling issue for detergent and water recovery from laundry wastewater.

Although the study objective is mentioned, some points should be considered before final acceptance:

  1. In Table 1, does NTU mean turbidity?
  2. In Table 2, add an appropriate reference for each analytical method.
  3. According to Fig. 1, why three opening ports are used?
  4. Why using different unit formats L/(m2 h bar); Lm-2h-1bar-1?!
  5. In the discussion parts, avoid extensive citation of previous studies.
  6. In Fig. 6, it’s still not clear the influent, effluent, and reuse ports.
  7. More figures are focusing on the flux and permeability; however, more data should be given to the pollution reduction, kinetics, removal rates, etc.
  8. Do not repeat the words in the title to the “keywords”.
  9. The last paragraph of Introduction should include the study objectives/procedures in brief
  10. The study should unveil the major gaps within the existing knowledge of the proposed model

Round 2

Reviewer 2 Report

The authors' responses to my comments are satisfactory.

This manuscript is a resubmission of an earlier submission. The following is a list of the peer review reports and author responses from that submission.

Round 1

Reviewer 1 Report

In do see a not acceptable overlap with “Bilad, M.R.; Mat Nawi, N.I.; Subramaniam, D.D.; Shamsuddin, N.; Khan, A.L.; Jaafar, J.; Nandiyanto, A.B.D. Low-Pressure 358 Submerged Membrane Filtration for Potential Reuse of Detergent and Water from Laundry Wastewater. J. Water Process Eng. 359 2020, 36, 101264 “ in Abstract, Introduction, Table 1.

From your sketch in Fig 1 it seems there is a certain crossflow. Can that be specified. What was the recirculation stream? Volume of tank with hollow fiber module?

Table 2 The results shown here can not easily (or at all) be connected on what is state in2.1 Preparation of feed waste water. Did you produce the “real” laundry waste water by yourself? What has been the composition after adding the detergent?

Is it correct that section 3.1 is addressing the results from gravity driven system (GDS)?

Line 185: … It is worth noting that the membrane compaction was highly reversible, judging from experiment repetitions… How do you measure the compaction and how can you know that is was reversible.

In figure 3: Do we see the unused detergent solution with 8 g/L detergent?

Line 198:  Can we really compare 8 g/L detergents with 2.5 mg/L (Table 2)?

In figure 4: Do we see the performance of the GDS with waste water described in Table 2?

Line 204: Final values provided in the text do NOT match the values in Figure 4A nor in Figure 4B.

Line 206: Which literature you mean precisely ? Even with a biofilm grown on such systems Derlon et al. 2014 achieved 9 L/m^2h. Not that bad.

Line 240: 16.4 % of 2.5 mg/L. You are nearly counting molecules. And better would be to show the rejection for the experiments shown in Figure 3.

I do not understand the story with phosphorus bound to large molecules. That is extremely sloppy. A little bit more of explanation would be nice.

How is the mechanism of color rejection?

Figure 6 is an idea but not well thought out. This system will stink after latest one week as you will have a high activity of micro organisms.

The authors did not invest interpretation of results. They even do not show the necessary ones, which would minimum be the rejection of COD and detergent of the water with detergent. I guess there will not be much rejection and then the authors would have to discuss that in more detail.

Reviewer 2 Report

Accept with Minor changes

This paper deals with a an important issue of water reuse in laundry  operations, using simple low-tech solutions. The writing and grammar is reasonable but there are places were an English language edit is necessary, eg. Abstract, line 211.

The introduction is sufficient although there is some unnecessary repetition (lines 83-88 could be cut/condensed).  

The methods are well-explained and thorough, although would be improved by adding the water quality characteristics of the “tap water”. Comment should be added to explain the influence and significance of “unsteady state” concentration in the wastewater reservoir for the experiments. For example, the volume of the reservoir and recirculation/feed flowrates should be noted, and the impact of changing concentration (in the reservoir) on the fundamental equations (1-3) should be explained (i.e. do these equations still hold given varying concentration?).

Unclear symbols at line 58: Cf and Cp ?

Section 3.2, lines 196-200. I am unconvinced of the assumed influence of fouling ? If that were the case then explain why this effect would not increase over time.

Lines 214-218: this paragraph is confusing. More extensive reporting of flux data would help. As such, paragraph at lines 220-226 feels like conjecture. Evidence should be presented.

Line 260-261: I found this result surprising ? What have others observed with respect to TDS ? Relate your results back to existing/previous literature.  

Discussion and conclusion: Your paper would benefit from a comment on how long the membrane will last and how it might be cleaned, to provide practical advice on implementation.